# Morphology, Microstructure, and Mechanical Properties of S32101 Duplex Stainless-Steel Joints in K-TIG Welding

**DOI:** 10.3390/ma15155432

**Published:** 2022-08-07

**Authors:** Shuwan Cui, Yunhe Yu, Fuyuan Tian, Shuwen Pang

**Affiliations:** School of Mechanical and Automotive Engineering, Guangxi University of Science and Technology, Liuzhou 545006, China

**Keywords:** K-TIG welding, geometry profile, Σ3 CSL grain boundary, random phase boundary, impact toughness

## Abstract

In this paper, the S32101 duplex stainless steel welded joints were produced by a K-TIG welding system. The weld geometry parameters under different welding speeds were analyzed by combining the morphological characteristics of the keyhole. The microstructure and impact toughness of the base metal and weld metal zone under different welding speeds were studied. The experiment results show that the welding speed has quite an effect on the geometry profile of the weld. In addition, the characteristic parameters of the keyhole can effectively predict the geometry profile of the weld. The test results prove that the microstructure, Σ3 coincidence site lattice grain boundary, and phase boundary of ferrite and austenite have an effect on the impact property of the weld metal zone. When the proportion of the austenite, Σ3 coincidence site lattice grain boundary and random phase boundary increased, the impact property of the weld metal zone also increased.

## 1. Introduction

Duplex stainless steel (DSS) is composed of two phases of ferrite and austenite with content close to 1:1, while the accepted ranges for austenite fraction are approximately 40–60%. The characteristics of the two-phase structure have the advantages of corrosion resistance, excellent toughness, higher strength, and low price [1,2,3,4]. It was often used in the manufacture of marine equipment such as offshore oil production platforms or large ships. Welding is the main technique for connecting ship structures. Welded joints were exposed to harsh environments for a long time, such as sea waves, wind loads and cold currents, which pose challenges to the performance of welded joints. The proportion of the two phases and the microstructure have an effect on the mechanical properties of the welded joint. The proportion and form of austenite mainly depend on the chemical composition and cooling rate. In traditional tungsten inert gas (TIG) welding, to form more austenite during the rapid cooling process, the filler metal with a nickel content of 2~4% higher than the base metal (BM) composition is usually used.

Westin et al. reported that filler metal with higher Ni content is usually used to further improve the proportion of austenite and ensure the excellent mechanical properties of the welded joints [5]. Mourad et al. reported that compared with electron beam welding (EBW), the welding heat input of TIG welding was relatively high, and the corresponding cooling rate was slower [6]. Therefore, the proportion of the two phases in the DSS TIG weld metal zone (WMZ) was closer to the BM. Pekkarinen et al. optimized the welding parameters of laser heat conduction welding [7]. By improving the cooling rate, the proportion of two phases in the WMZ were almost close to 1:1. When welding a certain thickness of welding plate, keyhole tungsten inert gas (K-TIG) welding does not require filler metal and groove processing. It can realize single-sided welding and double-sided molding [8,9]. Therefore, different welding speeds (WS) have a greater impact on the microstructure of the K-TIG WMZ. In previous experiments, the evolution of the microstructure of each area of the 32101DSS K-TIG welded joint has been discussed. However, the influence of WS on the geometry profile, microstructure, and impact toughness of welds was not analyzed. During the welding process, the keyhole penetrating the workpiece can be generated in the molten pool. The behavior and characteristic parameters of the keyhole can directly determine the geometry profile of the welded joint.

In this paper, the characteristic parameters of the keyhole were used to study the morphology of the K-TIG weld. The evolution laws of geometry profile, microstructure and impact toughness of the K-TIG welds under different WS conditions were analyzed.

## 2. Materials and Methods

### 2.1. Material and Welding Procedure

The BM grade of DSS selected in this study was UNS S32101. The S32101 DSS plates were cut into a rectangle with a dimension of 100 mm × 300 mm, and the thickness of the plates were 10.8 mm. The chemical compositions of the S32101 DSS plates are displayed in Figure 1. The K-TIG welding platform was built based on the vision system, as displayed in Figure 2. The vision system was mainly composed of the NSC1003 CCD camera produced by the French NIT company and a filter with a center wavelength of 810 nm, a bandwidth of 40 nm, and a light transparency of 80%. The acquisition frequency of the CCD camera was set to 14 f/s. The optical center of the lens of the CCD camera was in the same plane as the center line of the welding torch, the distance from the center of the weld on the back was 200 mm, and the angle to the horizontal plane was 35°. To produce high-grade welded joints, the front and back of the workpieces were mechanically cleaned before welding. Table 1 lists the experimental parameters in this investigation.

### 2.2. Keyhole Image Acquisition and Feature Extraction

The raw image of the keyhole on the back of the workpiece was obtained by the CCD camera, as shown in Figure 3. The X-axis represents the welding direction, and the Y-axis is perpendicular to the welding direction, as shown in Figure 3a. To accurately characterize the geometry of the keyhole, four characteristic parameters of the keyhole were extracted in this study, namely the length perpendicular to the WD (YA), the length along the WD (XA), the keyhole area (SA) and the ratio of XA and YA (XA/YA). The camera calibration experiment was carried out before extracting the features of the keyhole. The calibration coefficients obtained by calculation are: K*x* = 0.0299 mm/pixel (along the WD), K*y* = 0.04587 mm/pixel (perpendicular to the WD).

The brightness of the keyhole on the back of the workpiece in the image was quite different from the background. In order to effectively extract the four characteristic parameters, the keyhole edge should be extracted first. The effect diagram of each step in the keyhole edge extraction process is displayed in Figure 3b,c. The specific steps of keyhole edge extraction processing are as follows:(1)Combining the characteristics of the grayscale distribution of the keyhole grayscale image, use 5 × 5 disc-shaped structural elements for the corrosion operation;(2)The method of multi-level threshold segmentation was used to solve the binary image of the keyhole, and the edge points were obtained by line scanning. The specific threshold segmentation was realized by Formulas (1) and (2). *f* (*x*, *y*), *h* (*x*, *y*) and *g* (*x*, *y*) were the gray values at the coordinates (*x*, *y*) of the raw image and the image after two divisions. *T*_1_ = 240, and *T*_2_ is the threshold calculated by the Otsu algorithm for *h* (*x*, *y*). Finally, the ellipse is fitted based on the least square method, as shown in Figure 3c.
(1)$$h(x,y)=\left\{ \begin{array}{cc} 0,\; & f(x,y)<T1\\ f(x,y),\; & f(x,y)\geq T1 \end{array}\right.$$
(2)$$g(x,y)=\left\{ \begin{array}{c} 0,h(x,y)<T2\\ 1,h(x,y)\geq T2 \end{array}\right.$$

The geometric dimensions of the keyhole are displayed in Figure 4. YA was the largest dimension of the keyhole perpendicular to the WD. XA was the largest dimension of the keyhole in the WD. The geometric center position of the keyhole (*x*_0_, *y*_0_) was determined by YA and XA on the edge curve of the keyhole. The specific calculation formula is as follows: (3)$${\mathrm{YA}=y}_{2}-y_{1}$$
(4)$${\mathrm{XA}=x}_{4}-x_{3}$$
(5)$$x_{0}=\left(x_{1}{+x}_{2}\right)/2$$
(6)$$y_{0}=\left(y_{1}{+y}_{2}\right)/2$$
(7)SA=π×XA×YA/4
(8)$${\mathrm{XA}/\mathrm{YA}=(x}_{4}-x_{3}{)/(y}_{2}-y_{1})$$

### 2.3. Joint Characterization

The cross-sections of the welded joints perpendicular to the WD were cut by using wire-cut electrical discharge machining (WEDM) and polished with different types of sandpaper and 2 μm diamond paste. The Beraha etchant (30 mL H_2_O + 60 mL HCL + 1 g K_2_S_2_O_5_) was adopted to corrode the welded joint to observe the appearance of the weld [10].

The Charpy impact tests were performed on the WMZ according to ASTM A370 [11]. The dimensions of Charpy impact test specimens are displayed in Figure 5. The Charpy impact tests were repeated three times. The impact fracture morphologies of the WMZ under different WS were examined by using scanning electron microscopy (SEM). The Coincidence site lattice grain boundary (CSLGB) and phase boundary characteristics (PBC) of the WMZ were examined by the electron backscatter diffraction (EBSD) technique.

## 3. Results and Discussion

### 3.1. Geometry Profile of K-TIG Weld of S32101 Duplex Stainless Steel

The geometry profile on the front of the S32101 DSS K-TIG welds under different WS is shown in Figure 6. When the WS decreased from 5.0 to 4.0 mm/s, the three parameters of weld depth, weld width, and depth/width ratio of the weld all increased with the decrease of WS. In Figure 6, it was found that when the WS were 5.0 and 4.5 mm/s, the welds were not completely penetrated, and holes appeared. When the WS dropped from 4.0 to 3.0 mm/s, the workpieces were completely penetrated, and the weld depth remained unchanged. The weld depth/width ratio gradually decreased, and the weld width gradually increased with the decrease of WS. When the WS was 3 mm/s, the front of the weld seriously collapsed.

### 3.2. Dynamic Behavior and Characteristic Parameters of the Keyhole

#### 3.2.1. Dynamic Behavior of the Keyhole

To show the dynamic behavior of the keyhole, the keyhole images under different WS were arranged by rotating 90° counterclockwise, as shown in Table 2. The direction of the arrow was the WD, and the dimensions along the WD and perpendicular to the WD were 450 pixels and 250 pixels, respectively.

When the WS was 5.0~4.5 mm/s, the keyhole images on the back of the workpiece were not observed, and only the light emitted by the molten metal in the molten pool can be seen. When the WS was 4.0~3.0 mm/s, continuous and stable keyhole images were observed. When the thickness of the workpiece was constant, the keyhole was continuously and stably opened during the K-TIG welding process. However, the process window for the WS that can produce a continuous and stable keyhole was relatively narrow. It can be proved that the WS was the main factor affecting the welding quality.

In addition, under different WS, the time from the start of the arc to the formation of the keyhole through the workpiece was different. When the WS were 4.0 mm/s, 3.5 mm/s and 3.0 mm/s, the time from the arc starting to form the keyhole through the workpiece were 4.48 s, 2.92 s, and 2.02 s, respectively. It proved that when the thickness of the workpiece and other welding parameters were constant, the WS was lower; the shorter the time from the arc starting to the formation of the keyhole through the workpiece, the stronger the arc’s penetrating ability.

#### 3.2.2. Characteristic Parameters of the Keyhole

The keyhole images under different WS were edge-fitted, and the processed keyhole images (at a certain moment) are shown in Figure 7. The green curve was an ellipse fitted based on the least square method. When the WS was 3.0 mm/s, the geometric size of the green ellipse increased significantly, as shown in Figure 7c. When the WS were 4.0 and 3.5 mm/s, the geometric size of the green ellipse changed insignificantly, as shown in Figure 7a,b, respectively. In the K-TIG welding process, the keyhole images were collected according to time, so the value of the characteristic parameters, such as XA, YA, SA and XA/YA of the keyhole also changed with time, as shown in the Figure 8a–d. The blue line, red line and black line indicate that the WS were 4.0 m/s, 3.5 m/s and 3.0 m/s, respectively. When the WS were 4.0 m/s and 3.5 m/s, the value of XA, YA and SA of the keyhole change lessened, and they fluctuated within a certain range. However, when the WS was 3.0 m/s, the XA, YA, and SA of the keyhole significantly increased. In Figure 8d, it can be seen that under different WS, the value of XA/YA of the keyhole does not change significantly.

The average value of XA, YA, SA and XA/YA of the keyhole when the WS were 4.0~3.0 mm/s are listed in Table 3. When the WS dropped from 4.0 m/s to 3.0 m/s, the average value of XA, YA and SA of the keyhole gradually increased as the WS decreased. When the WS was 3.0 mm/s, the average value of the keyhole characteristic parameters changed most significantly. Compared with the WS of 4.0 mm/s, the average value of YA, XA and SA increased by 0.57 mm, 0.78 mm, and 1.86 mm^2^, respectively. The rates of change of YA, XA, and SA were 47.1%, 54.9%, and 139.8%, respectively. Although the XA/YA value of the keyhole was small, it can be proved that the keyhole was oval. When the WS was reduced from 4.0 to 3.0 mm/s, the value of XA/YA of the keyhole only increased by 6%.

Based on the above analysis, it proved that the keyhole was always open in the K-TIG welding process, which provided a channel for the escape of plasma gas. When the WS were 4.0~3.0 mm/s, the heat input gradually decreased as the WS increased. When the WS were faster (5.0 mm/s and 4.5 mm/s), the penetrating ability of the arc weakened, which eventually led to the failure to form a keyhole in the molten pool. The value of the characteristic parameters of the keyhole was 0. In the molten pool, the plasma gas cannot be ejected from the original channel, so the weldment is not completely penetrated, and holes are formed in the weld. When the WS dropped from 4.0 to 3.5 mm/s, the characteristic parameters of the keyhole changed little. The weld was completely penetrated, and the weld surface was well formed. Therefore, when the WS was relatively low (3.0 mm/s), the corresponding heat input was larger, and more metal was melted. At this time, the change rate of the average value of the keyhole characteristic parameters was relatively high. A large amount of liquid metal flowed down the inner wall of the keyhole, which made the front of the weld was seriously collapsed. The test results showed that the characteristic parameters of the keyhole can be characterized and describe the dynamic characteristics of the keyhole. In addition, the characteristic parameters of the keyhole also qualitatively explain the forming quality of the weld.

### 3.3. Impact Property

The Charpy impact test results of the BM and WMZ are shown in Table 4. When the WS was 4.0 mm/s, the average value of impact absorbed energy (IAE) of the WMZ was 142 J, which was 67.9% of the BM. When the WS was 3.5 mm/s, the average value of IAE of the WMZ was 172 J, which was 82.3% of the BM. The results showed that the average value of IAE of the BM is significantly higher than that of the WMZ, while the average value of IAE of the WMZ gradually decreased with the increase of WS. The fracture surface morphologies of the WMZ after the Charpy impact tests are shown in Figure 9. When the WS was 4.0 mm/s, there were a lot of dimples on the fracture surface of the WMZ, but there are obvious steps in the red area, as shown in Figure 9a. Therefore, this fracture mode was a mixed fracture mode. When the WS was 3.5 mm/s, there were more dimples and tear edges on the fracture surface of the WMZ, which indicated the fracture mode was a ductile fracture mode, as shown in Figure 9b. The impact fracture surface morphologies and Charpy impact test results showed that when the WS was 3.5 mm/s, the impact property of the WMZ was better.

### 3.4. Effect of Microstructure in the WMZ on Its Impact Toughness

#### 3.4.1. The Volume Fraction and Morphology of Austenite

Figure 10 display the microstructure of the BM and WMZ under different WS. The microstructure of the BM was mainly composed of austenite and ferrite, and the austenite was distributed in the ferrite matrix in a lath shape, as displayed in Figure 10a. The volume fraction of austenite in the BM was 48.1%. The austenite in the WMZ mainly exists in three forms: grain boundary austenite (GBA), intergranular austenite (IGA, and Widmanstätten austenite (WA), as displayed in Figure 10b,c. When the WS were 4.0 and 3.5 mm/s, the volume fractions of the austenite were 38.5% and 41.2%, respectively. Compared with BM, the volume fraction and morphology of austenite in the WMZ changed significantly. Faster WS (4.0 mm/s) results in a faster cooling rate, as there is insufficient time for part of austenite to precipitate in the WMZ. Therefore, the austenite volume fraction of the WMZ with a WS of 4.0 mm/s was significantly smaller. Zhang et al. and Cui et al. reported that the volume fraction of the austenite had an effect on the impact properties of the DSS welded joints [12,13]. The impact toughness of the WMZ increased with the increase of the austenite volume fraction. The volume fraction of austenite in the WMZ with a WS of 4.0 mm/s was lower than that in the WMZ with a WS of 3.5 mm/s, so the corresponding impact toughness was also weaker.

#### 3.4.2. Effect of CSLGB on Impact Toughness of Welded Joint

Σ (1 ≤ Σ ≤ 29) represents the proportion of coincidence site lattice (CSL) on the grain boundary (GB). When the value of Σ was smaller, the proportion of the coincident site was greater, and the GB structure was more orderly. At this time, the interface energy was lower, and the property of the GB was superior. Therefore, the Σ3 CSLGB had stronger fracture resistance than other grain boundaries. Σ3 CSLGB can hinder the shift of HAGB, and high-proportion, high-angle grain boundary (HAGB) can significantly reduce the ductile–brittle transition temperature [14,15,16]. Figure 11 shows the distribution characteristics of CSLGB in the BM and WMZ under different WS. It can be seen that the proportions of the Σ5 CSLGB, Σ7 CSLGB, Σ9 CSLGB, and Σ11 CSLGB in the BM and WMZ were low, almost close to 0. Therefore, this article mainly analyzed the influence of Σ3 CSLGB on the impact toughness of the welded joint. The proportion of the Σ3 CSLGB in the BM was 58.6%. When the WS were 4.0 and 3.5 mm/s, the proportions of the Σ3 CSLGB in the WMZ were 15.12% and 16.42%, respectively. By comparing the data of the Charpy impact tests of the BM and WMZ, it was proved that when the proportion of Σ3 CSLGB was larger, the corresponding impact toughness was also better. Therefore, it can be explained that the proportion of Σ3 CSLGB had a certain influence on the impact toughness of WMZ, which was consistent with the results of previous studies.

#### 3.4.3. Effect of Random Phase Boundary on Impact Toughness of Welded Joint

In DSS, the interface properties of ferrite and austenite also have a significant impact on the impact toughness of the material. Piñol-Juez [17] and Patra [18] pointed out that the interface between ferrite and austenite following the Kurdjumov–Sachs (K–S) orientation was less prone to sliding than the boundaries of other orientation interfaces. The boundary of the K–S interface was easier to form cracks than the boundary of other orientation interfaces. However, the precise K–S orientation relationship was almost non-existent. Therefore, based on the deviation of K–S orientation, the phase boundary of ferrite and austenite in DSS is divided into two types [19]. One is the phase boundary where the misorientation angle is less than 6°, called the special phase boundary, and the other is the phase boundary where the misorientation angle is greater than 6°, which is called the random phase boundary.

Figure 12 displays the orientations of the phase interfaces between the ferrite and austenite in the WMZ and the BM. The blue phase is the ferrite phase; the white phase is the austenite phase. The yellow line represents the special phase boundary, and the red line represents the random phase boundary. When the WS are 4.0 mm/s and 3.5 mm/s, the orientations of the phase interfaces between ferrite and austenite in the WMZ are shown in Figure 12a,b. Figure 12c shows the orientations of the phase interfaces between ferrite and austenite in the BM. Figure 12d displays the proportion of the random phase boundary in each area. The proportion of the random phase boundary in the BM was 70.9%. When the WS were 4.0 and 3.5 mm/s, the proportions of the random phase boundary of ferrite and austenite in the WMZ were 27.2% and 29.7%, respectively. The test results show that the random phase boundary has an effect on the impact toughness of the S32101 DSS WMZ. When the proportion of random phase boundaries increased, the toughness of WMZ gradually increased, which was consistent with the conclusions of Zhang et al. [20].

## 4. Conclusions

The S32101 DSS welded joints were produced by applying a K-TIG welding system under different WS. When the WS were 3.5 mm/s and 4.0 mm/s, the weld morphologies were better.The keyhole was oval, and the characteristic parameters of the keyhole were dynamically changed during K-TIG welding. When the WS decreased, XA, YA, and SA of the keyhole changed significantly, which proved that the characteristic parameters of the keyhole can indirectly indicate the forming quality of the weld.When the WS was 3.5 mm/s, the impact toughness of the WMZ was better. The microstructure, GB orientation characteristics, and phase boundary have a certain influence on the impact toughness of the WMZ. When the proportion of the austenite, Σ3 CSLGB and random phase boundary increased, the impact toughness of the WMZ also increased.

## Figures and Tables

**Figure 1 materials-15-05432-f001:**
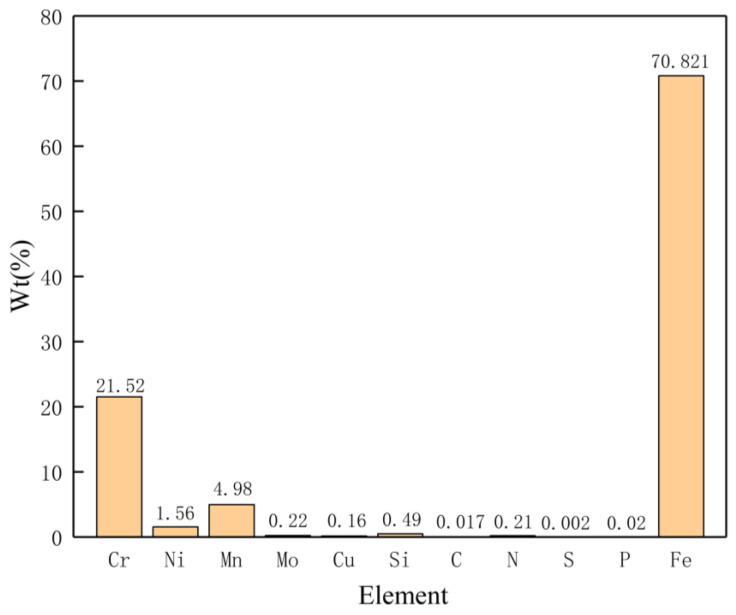
Chemical composition of the S32101 DSS plate.

**Figure 2 materials-15-05432-f002:**
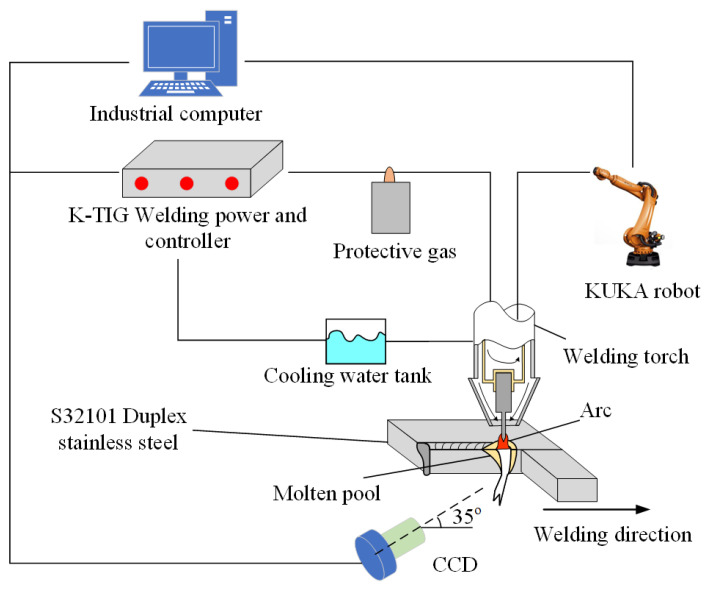
The K-TIG welding platform.

**Figure 3 materials-15-05432-f003:**
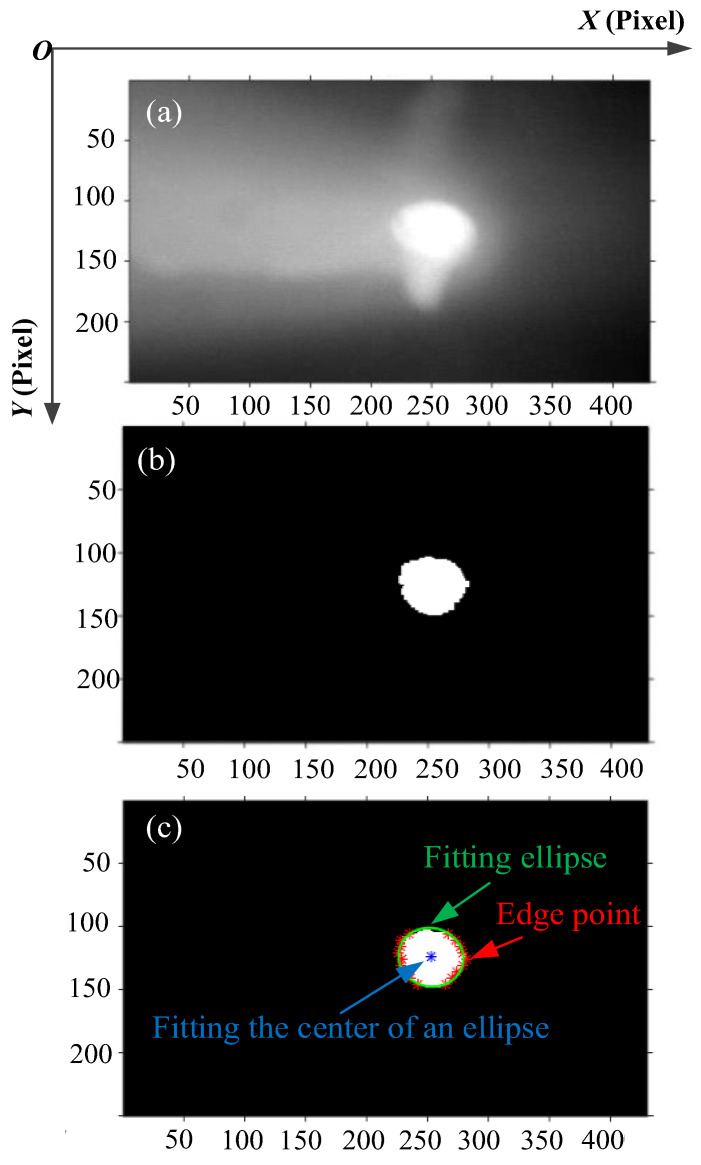
Keyhole edge extraction effect diagram: (**a**) raw image; (**b**) two-level threshold segmentation; (**c**) elliptical curves fitting.

**Figure 4 materials-15-05432-f004:**
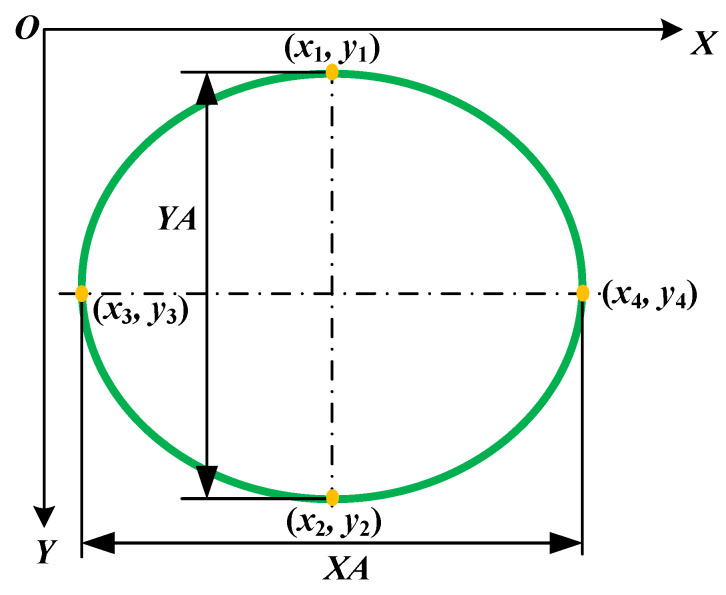
The geometric dimensions of the keyhole size.

**Figure 5 materials-15-05432-f005:**
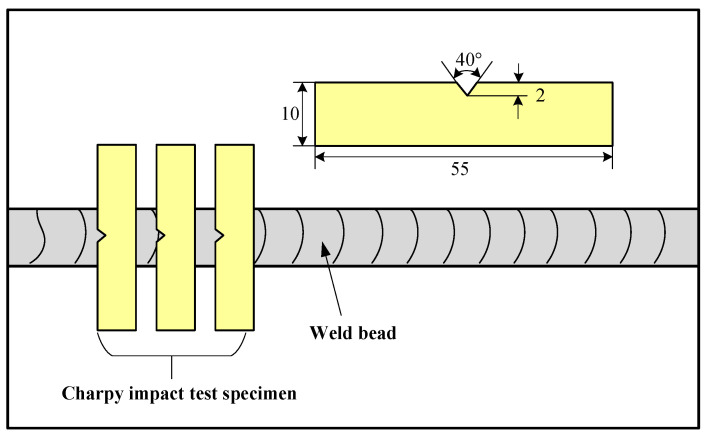
Dimensions of the Charpy impact test specimens.

**Figure 6 materials-15-05432-f006:**
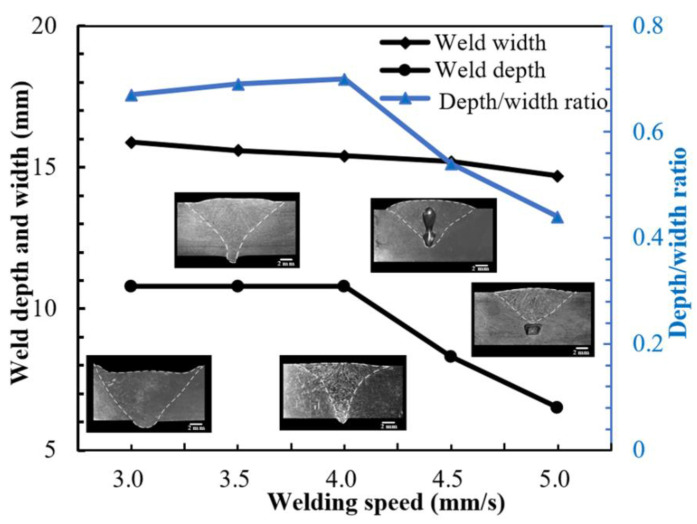
Geometry profile on the front of the S32101 duplex stainless-steel K-TIG welds.

**Figure 7 materials-15-05432-f007:**
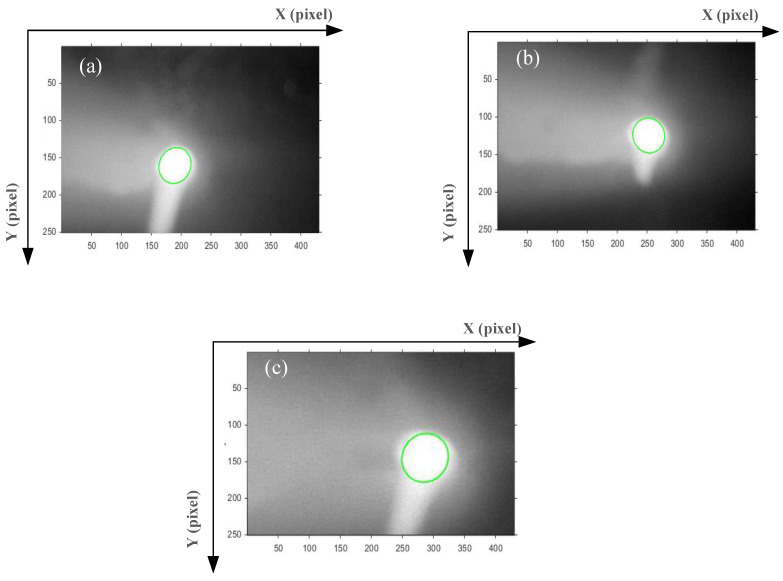
Images after edge fitting of the keyhole under different welding speeds: (**a**) 4.0 mm/s, (**b**) 3.5 mm/s, (**c**) 3.0 mm/s.

**Figure 8 materials-15-05432-f008:**
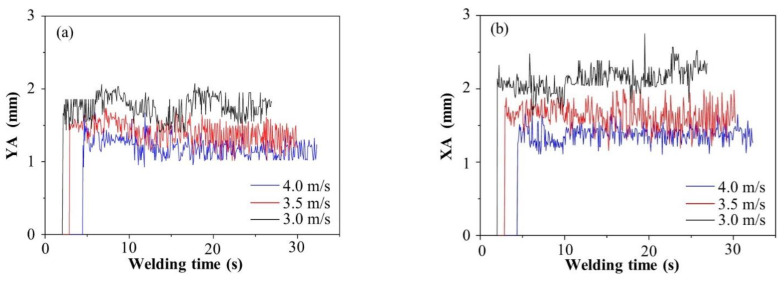
Characteristic parameters of keyhole exit under different welding speeds: (**a**) YA, (**b**) XA, (**c**) SA, (**d**) XA/YA.

**Figure 9 materials-15-05432-f009:**
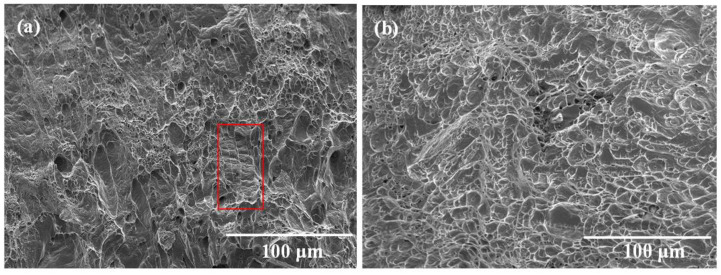
Fracture surface morphologies of the WM after Charpy impact tests under different welding speed: (**a**) 4.0 mm/s, (**b**) 3.5 mm/s.

**Figure 10 materials-15-05432-f010:**
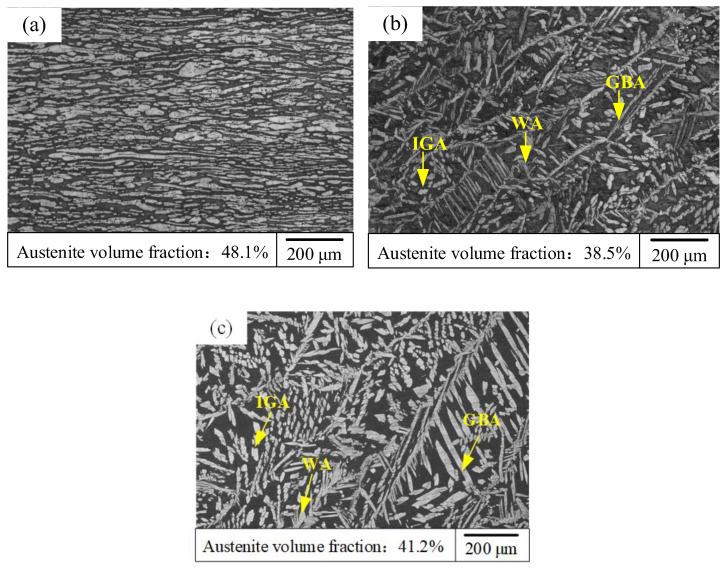
Microstructure of the BM and WM: WM under different welding speeds (Unit: mm/s): (**a**) BM, (**b**) 4.0, (**c**) 3.5.

**Figure 11 materials-15-05432-f011:**
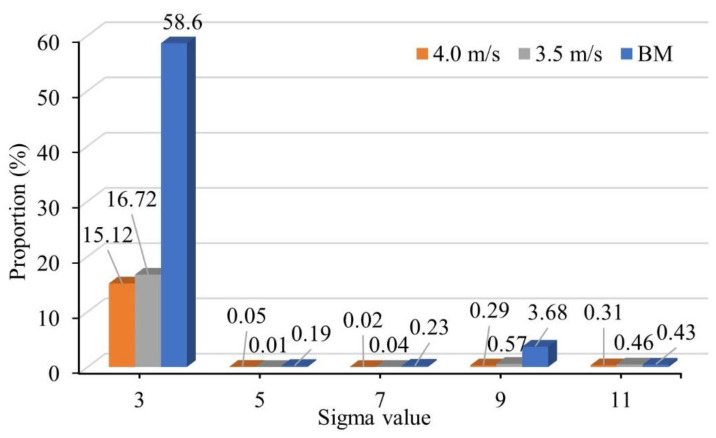
Distribution characteristics of CSL grain boundaries in the BM and WM under different welding speeds.

**Figure 12 materials-15-05432-f012:**
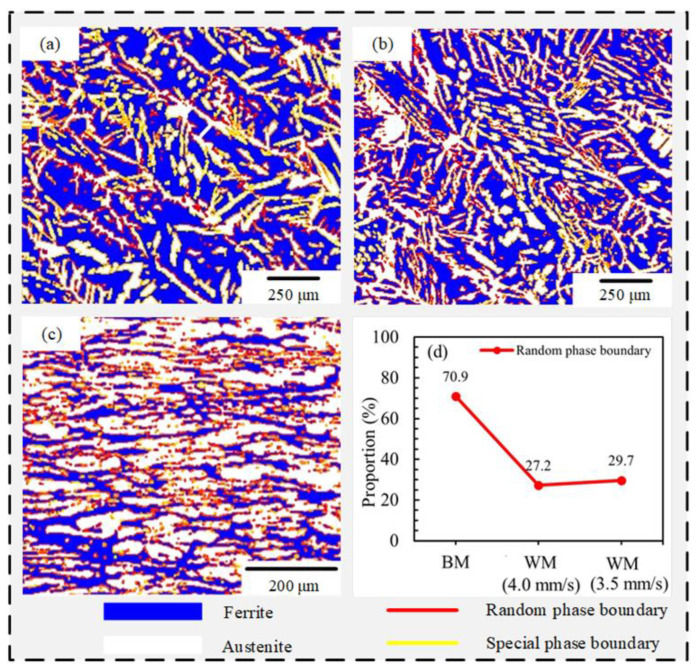
Orientations of the phase interfaces between ferrite and austenite in the WM and the BM. WM: (**a**) 4.0 mm/s, (**b**) 3.5 mm/s, (**c**) BM, (**d**) the proportion of the random phase boundary.

**Table 1 materials-15-05432-t001:** Welding parameters.

Test	Welding Current	Welding Speed	Flow Rate of Argon
	(A)	(mm/s)	L/min
1	530	5.0	20
2	530	4.5	20
3	530	4.0	20
4	530	3.5	20
5	530	3.0	20

**Table 2 materials-15-05432-t002:** Keyhole image on the back of the workpiece under different welding speeds.

Welding Speed (mm/s)	Keyhole Image
5.0	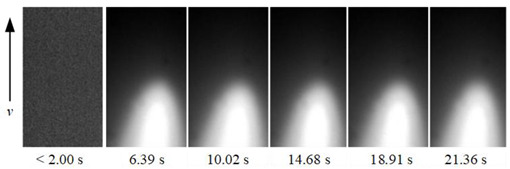
4.5	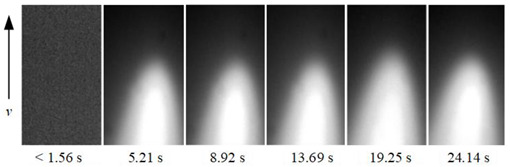
4.0	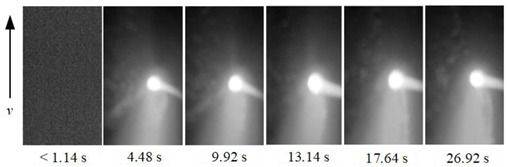
3.5	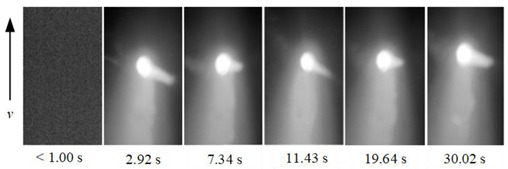
3.0	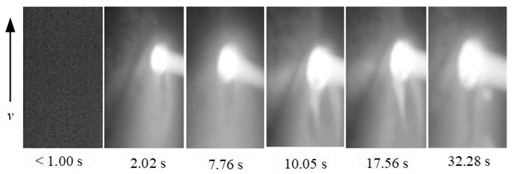

**Table 3 materials-15-05432-t003:** Mean value of keyhole characteristic parameters under different welding speeds.

Keyhole Parameters	Welding Speed (mm/s)			Rate of Change (4.0 mm/s → 3.0 mm/s, %)
	4.0	3.5	3.0	
YA (mm)	1.21	1.32	1.78	47.1
XA (mm)	1.42	1.62	2.20	54.9
SA (mm^2^)	1.33	1.64	3.19	139.8
XA/YA	1.16	1.17	1.23	6.0

**Table 4 materials-15-05432-t004:** Absorbed energy of the BM and WM.

	Welding Speed (mm/s)	Absorbed Energy, Akv (J) (20 °C)			
		1	2	3	Mean Value
WM	4.0	148	146	133	142
	3.5	166	172	179	172
BM	/	205	215	208	209

## Data Availability

Not applicable.

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
