# Peer review of "Morphology, Microstructure, and Mechanical Properties of S32101 Duplex Stainless-Steel Joints in K-TIG Welding"

_materials, 2022, doi:10.3390/ma15155432_

Round 1

Reviewer 1 Report

This paper presented excellent experimental results of UNS S32101 duplex stainless steel samples and micro/macro-structures and the application of K-TIG welding process.

All the results are very well prepared form. However, it needs minor revisions:

# on page 01, line 37: …….electron beam welding (EBW)…….;

# on page 01, lines 39,40: change Weld Metal Zone(WMZ) to Fusion Zone(FZ) – “this is for all text”

# on page 03:   lines 74, 75, 76: The sentences are confused

#on page 03: Fig. 2a is not cited in the text.

# on page 07:   Figures 6a, 6b and 6c must be cited in the text.

# on page 08:   Figures 7a, 7b, 7c and 7d must be cited in the text.

# on page 10, line 211: ….of the fusion zone are shown in Fig 8…..

#on page 10:  Fig. 8b is not cited in the text.

# on page 11:   put figures a, b and c in order.

#on page 12:  Fig. 11c is not cited in the text.

#on page 13, line 289:  Fig. 11d is not cited. on page 11:   put figures a, b and c in order.

#on page 12:  Fig. 11c is not cited in the text.

#on page 13, line 289:  Fig. 11d is not cited.

Author Response

We revised the paper according to your comments, the details are in the attachment.

Reviewer 2 Report

It would be interesting if the authors gave the influence of the change in the welding speed on the change in the initiation energy and the propagation energy when determining the Charpy impact toughness.

Author Response

(The authors gave the same response as above.)

Reviewer 3 Report

This manuscript is nearly ready for publication, only a few minor revisions are needed:

1)     The first time you use TIG and K-TIG, please spell out the acronym.

2)     In your introduction, please give a range for what is a good amount of austenite.

3)     Why is the first acronym you use for electron beam welding LBW? Shouldn’t it be EBW?

Author Response

(The authors gave the same response as above.)
